# Problems of Tourist Mobility in Remote Areas of Natural Value—The Case of the Hajnowka Poviat in Poland and the Zaoneshye Region in Russia

**Elzbieta Szymanska** 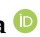

Department of Marketing and Tourism, Bialystok University of Technology, 15-351 Bialystok, Poland; e.szymanska@pb.edu.pl

**Abstract:** Tourist destinations are often inaccessible due to mobility problems. The purpose of this paper is to identify the mobility problems of tourist destinations in remote areas of natural value. The research was carried out in the following two tourist destinations with the above-mentioned values: in the Zaoneshye Region in Russia and the Hajnówka Poviat, which includes the priceless resources of the Polish part of the Białowieża Forest. The research was conducted using a survey method. Respondents could download the survey questionnaire onto their mobile devices (smartphone, tablet) by scanning a QR code or provide their answers to the questionnaire on paper or to an interviewer, who recorded them in an electronic version. The respondent group consisted of tourists visiting both regions for tourism purposes. The survey was carried out between 2019 and 2021. The results showed that the car is the preferred means of transport in both regions, and that road works are somewhat or completely necessary. Additionally, tourists in the Hajnowka Poviat travel a lot on foot or by bicycle, as there are more cycling and pedestrian paths available. In contrast, tourists visiting the Zaoneshye Region suggest providing more facilities for tourism and better and more efficient communication.

**Keywords:** tourism destination; natural value areas; rural areas; mobility problems

## 1. Introduction

One of the most important factors contributing to the development of tourism in natural value destinations in remote areas is the improvement of mobility and accessibility. Among the many problems of these areas, there are shortcomings in the transport sector. The lack or limited access to means of transport is essential to limit access to tourist attractions. We may describe this situation as "transport poverty". Poor transport accessibility is often the result of dispersed settlement structures, which makes it difficult to ensure an efficient public transport system. Mobility in remote natural areas, therefore, depends mainly on owning a private car (Soder and Peer 2018), which contributes to the degradation of valuable natural resources. On the other hand, investments in public transport infrastructure are aimed at limiting the negative impact on the natural environment and reducing spatial and social inequalities by improving tourists' access to these attractions, and residents to workplaces and other forms of activity (Oviedo et al. 2019).

The paper describes the current mobility situation in the Zaoneshye Region and the Hajnówka Poviat, and subsequently points out the associated mobility needs of tourists. Following that, there is a description of possible disparities between the current situation and newly revealed needs. The research will provide some recommendations on how to progress with work on mobility in both regions, complemented by a brief summary at the end. The Zaoneshye Region and the Hajnówka Poviat are partners in an EU project that aims to shed light on and improve mobility and accessibility in sparsely populated areas. The project is financed by the Interreg Baltic Sea Region programme. The project was launched in January 2019 and continued until September 2021. The project involves twelve partners from nine countries. The project aims to elucidate the challenges related to

mobility and ease of transport in sparsely populated areas and to investigate to what extent changes in these factors will affect settlement patterns and tourism attractiveness. The study provides a basis towards a better understanding of the current mobility situation in order to identify potential areas of improvement within the current provision. The totality of the survey is emphasised in the analysis, discussion and conclusion.

The main objective of the research is to identify the problems of tourism destination mobility in remote areas of natural value on the basis of research carried out in Zaoneshye District and Hajnówka Poviat.

### 1.1. Mobility Problems in the Literature

Mobility, according to the definition proposed by Szołtysek (2011), can be defined as the tendency to change one's place of residence or place of work. Therefore, mobility is associated with the crossing of an area and with various forms of mobility treated as the result of certain conditions and processes, without the possibility to influence their outcome (Kruszyna 2010). Rural and peripheral areas suffer from accessibility and mobility problems that challenge their livability and development potential (Vitale Brovarone 2022). The problem of transport accessibility concerns especially seniors (Ahern and Hine 2012; Plazinić and Jović 2018). In order to address these challenges, the spatial, social, cultural and economic components of accessibility need to be recognized and addressed with comprehensive actions that involve actors from different sectors at different scales (Atasoy et al. 2015; Clotteau 2014). Researchers point to numerous mobility problems, constantly looking for answers to the question—why do some regions decline and others develop (Li et al. 2019)? Scientists point to different causes, which was widely discussed at scientific conferences, such as the Conference CIVITAS FORUM (2018), where scientists blamed, among other things, the lack of cooperation in transport planning. Mobility as a Service (MaaS) multidisciplinary concept (Esztergár-Kiss and Kerényi 2019) was developed on the basis of various studies. Another concept is inclusive transport (Jeekel 2019). Also in the European Union, work to improve mobility was consolidated and the European Mobility Management Platform was created (European Platform on Mobility Management-EPOMM 2013). In addition, the third priority of the Interreg Baltic Sea Program (https://www.interregeurope.eu/ accessed on 30 August 2022) is dedicated to sustainable transport, and changes are made in individual regions of the European Union in line with the principles of sustainable transport or smart mobility (Gross-Fengels and Fromhold-Eisebith 2018).

From a tourist's perspective, developed public transport and road infrastructure are becoming increasingly important in rural areas. In fact, from a tourist's perspective, transport infrastructure is a major determinant of a region's accessibility. One of the most important points is the extent to which land-use and transport systems make it possible to visit tourist destinations by means of transport (Geurs and van Wee 2004). Multimodal planning establishes communities where walking, cycling and public transport are possible. This provides various benefits to tourists. Current trends include increasing demand for non-car travel options in rural areas, safety concerns and growing tourism industries (Litman 2019) The expected changes in the mobility of tourists take into account the principles of sustainable tourism (Scuttari and Isetti 2019).

MARA aims to validate the actual mobility needs of residents and tourists with the current mobility offers. The project aims to increase the capacity of regional and local transport actors to address multifaceted mobility needs by improving the existing services, as well as developing and testing innovative sustainable mobility solutions for remote areas. Finally, the project will integrate its improved or new mobility approaches in remote areas into regional spatial and mobility development plans. The territory of the MARA project includes a part of the Republic of Karelia (Russia). The focus area includes three rural settlements located on the Zaonezhsky peninsula, which is a part of the Medvezh'egorsk municipal district, located north of the regional capital Petrozavodsk.

Rural mobility in the Baltic Sea region faces several common challenges.

*1.2. The Natural Valuable Tourism Destination Mobility Problems in the Literature*

Swarbrooke (1995) proposed a typology of tourist attractions and grouped them in three categories, entertainment, heritage and emotions. Areas of natural value fall into the first two categories. Mobility in the context of tourism has repeatedly been the subject of academic research. The main findings of Zamparini and Vergori (2021) show that mobility at home, the use of a friendly mode of transport to reach a destination and the choice of a static holiday in places associated with the sea, sun and sand are the most relevant variables that positively influence environmentally friendly mobility. In addition, improved infrastructure and more appropriate mobility policies and strategies can influence more sustainable transport choices of visitors and residents. (Diskinson and Lumsdon 2011). The authors describe different types of slow tourism, namely walking tourism, cycling tourism, bus and coach tourism, train tourism, water-based travelling. They emphasise that slow tourism is more environmentally friendly. A book on sustainable transport in natural and protected areas, (ed. Orsi 2015) in which numerous authors address similar issues, should be considered a very valuable publication. Of particular note is the chapter that presents the sustainability potential of various transport modes in natural settings. The authors conclude that a sustainable transportation system guarantees the satisfaction of multiple environmental, social and economic requisites across space and over time.

Results on the mobility of residents in the Hajnowka Poviat were presented in 2022 (Szymanska and Koloszko-Chomentowska 2022). This research showed a wide range of opinions on public transport. The high rate of tourists' lack of opinion on this subject is precisely due to problems with public transport accessibility, which forces tourists to rely on private means of transport. Such responses are, on the one hand, a limitation of the survey, but on the other hand, they indirectly show a serious mobility problem.

Page and Connell (2020), in turn, undertake a systematisation of tourism issues in their book, including transporting the tourist (pp. 161–86) and rural tourism (pp. 466–83). Cohen et al. also undertake a discussion of mobility issues in tourism (Cohen et al. 2014).

Shen et al. (2019) proves that a better geographic location with greater accessibility is usually an advantage for rural tourism market expansion, as urban residents are still the main target market for rural tourism. Kirilenko et al. (2019) takes a similar approach. According to Sharav et al. (2019), the development of railways contributes to increasing the level of tourist penetration of destinations. Activities in line with the principles of sustainable tourism are key to its development, favouring naturally valuable tourist destinations (Borkowska-Niszczota et al. 2014). The organisation and functioning of clusters support tourism development, including tourist mobility (Sahakyan et al. 2019). Variables for the evaluation of tourist behaviour (accommodation, means of transport, frequency of visits, travel group) depend on the type of settlement unit and its location in a settlement network (Bartosiewicz and Pielesiak 2019). Descriptive statistics for the analysis of tourist length of stay in rural areas were based on the following three variants proposed by Więckowski et al. (2014): short term, medium term, long term. Innovative solutions for developing sustainable transport and improving tourist accessibility are very important (Szymańska et al. 2021). In Italy, Coppola et al. (2020) proposed the development of an Italian National Tourism Mobility Plan, which identifies one of the key drivers of investment in accessibility. For this purpose, they have developed a planning support system (PSS) with the aim of identifying investments that seek to close the accessibility gap of national tourist sites from the main airports, ports and railway stations (i.e., the 'access gates' to a country), either on the road network or using public transport services.

Based on the analysis of the above-mentioned literature, the research conducted focused on the following research problems: assessing the current state and prospects of infrastructure development in the context of the accessibility of individual tourist attractions and from the perspective of different means of transport.

**Hypothesis 1 (H1).** *States that the most popular form of travel in natural value remote areas is road transport.*

*1.3. Characteristics of the Research Areas*

The research was carried out in parallel in two areas from the countries of the Baltic Sea basin. Both research areas share the following characteristics:

- Remoteness from economic centres;
- Peripherality;
- Mobility and accessibility problems;
- Tourist attractiveness, consisting of valuable natural assets.

The area of interest within the MARA project in the Republic of Karelia (RUSSIA) is the Zaonezhye area, which includes the large Zaonezhsky peninsula and the adjacent archipelago of the Kizhi skerries (about 500 islands), with an area of 560 km$^2$. Its northern boundary runs through a natural watershed to the north of the Zaonezhsky peninsula. It is a unique historical and cultural complex with a historically formed settlement system, which administratively belongs to the Medvezh'egorsk municipal district of the Republic of Karelia. A large number of shallow rivers and deep-water lakes characterise the relief of Zaonezhye. Frequently, there are alternating elongated bays, lakes and long narrow rocky ridges, with a strict orientation from north-west to south-east. The historical transport routes for the Zaonezhye area include inland waterways (Lake Onega). Residents of Zaonezhye have created a particular type of boat named the "kizhanka", which is popular on Lake Onega even to this day. The road network is poorly developed due to the complex relief and water obstacles.

There are three rural settlements on the territory of the Zaonezhsky peninsula (Velikaya Guba, Tolvuya and Shun'ga). Each of them consists of several small villages (about 90 in total); some of them are inhabited only during the summer season. The total population of the peninsula is around 3500. The population has been declining for more than 10 years. Another trend is the ageing of the population; young people are leaving mainly for the district centre Medvezh'egorsk and the regional capital Petrozavodsk.

The territory of Zaonezhye is famous for its magnificent nature, historical and architectonic monuments, the pearl of which, the Kizhi Island, is a UNESCO monument. In 1966, the State Historical, Architectural and Ethnographic Museum-Reserve "Kizhi" was established. In 1990, Kizhi was inscribed on the UNESCO World Heritage List.

The Kizhi State Nature Reserve under the jurisdiction of the federation includes the protection zone of the Kizhi Museum-Reserve. The protected area of the Kizhi Museum-Reserve is located on an area of 50,000 hectares and has been established to protect rare species of flora and fauna and waterfowl breeding sites. The Museum-Reserve is also located in close proximity to the planned Kizhi Skerries National Park (the second option is the Zaonezhsky Nature Park) with an area of 115,000 hectares, whose main objective is to preserve the natural and cultural values of the northern part of Zaonezhye.

Despite the high attractiveness of the region, accessibility and communication are very difficult and include the following options:

- Water transport between Petrozavodsk and Zaonezhie;
- A bus service runs between Velikaya Guba and Medvezhyegorsk only once a day;
- The journey takes about five hours one way.

It must, therefore, be recognised that the region lies a long way from an economic centre, such as Petrozavodsk, and access to it is extremely difficult.

The region that represents Poland was the Hajnowka Poviat, which covers one of the most valuable natural areas in Europe, the Białowieża Forest with the Białowieża Forest Reserve. The Hajnowka region is characterised by a low percentage of county and municipal road density. The length of hardened surfaces is 39.5 km per 1 km$^2$. This is considerably less than the corresponding indicators for the Podlaskie Voivodeship (65.1 km per 1 km$^2$) and entirety of Poland (94.1 km per 1 km$^2$). The low road density in the Hajnowka region is mainly due to the large area of forest complexes (50.6%) and the low population density, which is 27 persons per 1 km$^2$ compared to 124 persons per 1 km$^2$ in Poland (US 2019; CSO (Central Statistical Office of Poland) 2019). Under these conditions, the organisation of public transport is quite a challenge, especially as the county is home to around 150,000 inhabitants in 244 localities.

Comparing the studied regions, it is possible to point out the following characteristics of both areas:

(a)  Remoteness from economic centres;
(b)  Peripherality;
(c)  Mobility (accessibility) problems;
(d)  Attractiveness for tourists in terms of valuable natural assets.

Both regions are remote areas and have a wealth of naturally valuable tourist attractions. Due to the location of both regions, improving mobility is a major challenge.

### 1.4. Research Gap and Expected Contribution to Business Practice

The indicated research gap shows the scarcity (lack) of research on tourist mobility in terms of innovation opportunities. Meanwhile, in the study areas, these are of a pioneering nature. Another novelty is the opportunity to compare such different research areas (Poland and Russia), although with similar natural valuable values. The scientific contribution of the study is the development of a research tool to study the mobility of tourists in different regions with similar tourism values. The expected contribution to business practice is a recommendation for the inclusion of the obtained results in the development strategies of the studied regions.

### 2. Methods and Materials

Due to the diverse social and political situation in both countries, both the research and the research procedures were adapted to the existing conditions and limitations. In addition, during the course of the project, the coronavirus pandemic began, which hampered the research process. However, despite the difficulties in both cases, every effort was made to achieve the set objectives.

The following formula was used in calculating the minimum sample size for an infinite population, following guidance from the Statistical Office (https://www.statystyka.az.pl/dobor/kalkulator-wielkosci-proby.php, accessed on 18 November 2020):

$$Nmin = z^2 P(1 - P)/e^2$$

The symbols used in the formula are as follows:

$P$ is the estimated fraction size—infinite fraction size;

$z$ is the value resulting from the assumed significance level ($\alpha$), calculated using the cumulative distribution function of a normal distribution;

$e$ is the maximum estimation error.

In the conducted empirical studies, the following assumptions were made for the infinite population; when the researcher is not able to estimate the size of the fraction $P$, its value should be set at 50% by default. Accordingly, the following assumptions were made:

• Estimated fraction size $P$ = 50%;
• Significance level $z$ = 5% (0.05);
• Acceptable error $e$ = 0.5 (5%).

The sample size calculation allows the minimum sample size to be determined and the resulting figure should be a natural number, which under the given assumptions is 384 units (respondents). The study used simple random sampling. The sample size for an infinite population (here: tourists) is calculated for quantitative research and has wide applicability in statistical research. In the survey part, the empirical analysis of the results and their prioritisation was based on the respondents' indications. The indicator that differentiated the level of impact of individual factors was the number of respondents' indications for a given factor and its level. In this way, structure indicators were calculated. Here, the structure indicator means the number of statistical units characterised by the n-th variant of a given characteristic, in relation to the number of all statistical units surveyed, and indicates the share of statistical units that possess the n-th variant of the characteristic in the entire surveyed population; it is usually presented as a percentage share. In the assessment of mobility needs and proposed innovative solutions, structure indicators presented in the form of a percentage share were used, on the basis of which the factors were prioritised, starting from a value of 1, indicating the lowest position in the hierarchy, to a value of 5, indicating the highest position in the hierarchy.

The research on the mobility needs of tourists in the Hajnowka Poviat was conducted using a survey method. The survey in Poland was conducted between 2019 and 2021. Two research methods were used, F2F (face-to-face) and CAWI. Respondents and interviewers were able to download the survey questionnaire to their mobile devices (smartphone, tablet) by scanning the QR code. They had access to a paper version of the questionnaire to provide their answers in the questionnaire or to the interviewer who would record them electronically. The survey conducted in Russia was translated and partly adapted from a questionnaire developed by the Bialystok University of Technology in Poland (in Hajnowka Poviat). The respondent group consisted of tourists visiting both regions for tourism purposes. The distribution of a representative sample of Polish tourists is N = 421, while there were 390 respondents in Russia. The questionnaire contained semi-open questions, with a developed set of multiple- or single-choice answers and evaluation questions. The evaluation question used a five-point Likert scale (Poland). The survey questionnaire consisted of the following three parts: preamble, questions concerning the research problem, and respondent specifications. Some questions and answer options concerning means of transport that do not occur in Karelia, i.e., questions on boats, planes and railways, were removed.

The collection and processing of information on the accessibility study in Zaonezhye as part of the MARA project was organised in two stages. The first stage of the work was carried out in summer 2019, together with the Kizhi Museum-Reserve and the "Kizhi Ozherel'e" (necklace) and "Karelia Excursion Bureau" travel companies, as part of a study to determine the motivations of tourists from different regions of Russia and foreign countries to visit Kizhi Island. The second phase was organised in the summer of 2020, when the research expedition aimed to survey local residents and tourists to determine motivations for visiting the sites and the accessibility of the area, and to compile the resulting data. In 2020, the interviewers were interested in the purpose of the trip, the availability of transport services, and the services needed in the remote areas of Zaonezhye. The survey was conducted in the village of Oyatevshchina, Velikaya Guba and on the islands of the Kizhi skerries. This is the first time such work has been carried out in the territory of Zaonezhye in the last few decades, The field phase of the study was organised between June and September by the Centre of Social Tourism Development, at the request of the Tourist Information Centre of the Republic of Karelia. The NGO "Zaonezhskaya Izba" and the Sailing Federation of Karelia were involved in the collection of information. Tool development, data processing and analysis (data entry, data processing and analysis, report preparation) were carried out by sociologist A.G. Chukhareva (Sociological Laboratory of PetrSU). The obtained quantitative data was processed and analysed in SPSS between October and November 2020.

The description of respondents was carried out in a slightly different way due to the rules of the survey directly (Poland) and through contacts of state institutions (Russia). The survey showed that the main purpose of visiting Zaonezhye for the majority of respondents (74.7%) was tourism, while one in ten respondents (8.8%) were a local resident. Of the tourists surveyed in Poland, 46.1% were men and 53.9% were women. Other variables were not examined, so there is no need to provide detailed characteristics of the respondents in Poland.

Due to the coronavirus pandemic and the temporary ban on visiting Kizhi Island in June, the majority of respondents were locals and dacha residents from the villages of Oyatevshchina, Ersenevo, Boyarshchina, Sychi, Yamka, Sennaya Guba, Potanevshchina, Zharnikovo, Korba and Volkostrov. However, already in July and early September, the questions were already answered by tourists from different parts of Russia from Apatity to Bryansk and Belgorod, from Kaliningrad to Perm and Orenburg. In total, the respondents came from 84 cities and regions (except Karelia). The largest influx of tourists was recorded from Moscow (20.7%), Petrozavodsk (19.3%) and St. Petersburg (13.6%). Only two foreign visitors were recorded, from Kiev (Ukraine) and Brest (Belarus).

In general, it should be considered that both groups of respondents constituted a representative sample of the population of tourists who visited both destinations in the analysed period.

## 3. Results of the Survey: Tourist Mobility Problems in the Hajnowka Poviat and the Zaonezhye Region

An examination of the respondents' answers showed that the general results regarding mobility problems in the two surveyed nature conservation areas are similar, but due to different political and legal conditions and climatic conditions, the specific expectations of tourists differ.

The majority of respondents (92.3%) use road transport when travelling to Zaonezhye, and 9.4% use water transport. They visit the territory for a varying number of days, depending on the purpose of their trip. Other modes of transport, such as bicycle (1.6%), train (1.5%) and aeroplane (1%), had a minor share.

The mobility needs of tourists were measured using the structure indicator (%) regarding Zaonezhye and Kizhi Island and can be summarised as follows:

- There are no alternative transport or excursion routes to the water transport to Kizhi Island and Kizhi skerries;
- Tourism infrastructure along the route R-17 Medvezh'egorsk–Velikaya Guba is not developed.

A survey among tourists regarding the assessment of the condition of infrastructure in Hajnowka County indicates that in categories as access to public transport (57%), frequency of public transport (64%), cost of public transport tickets (71%), availability of information on transport (64%), facilities for the disabled in public transport (77%), the vast majority of respondents do not have a precise opinion, neither positive nor negative, which is due to the fact that most of them do not use local transport. When it comes to assessing the technical state of the transport infrastructure, the majority of respondents (57%) rate the technical state of the transport infrastructure as rather positive and very positive; similarly, respondents rate the safety of their journeys as rather positive and very positive (Figure 1).

The assessment regarding problems with the introduction of changes in the functioning of bus and rail transport and road infrastructure, including bike paths in the Hajnowka Poviat, shows that the vast majority of tourists do not have an opinion on the need for changes in the functioning of local bus and rail transport.

The median, mode, arithmetic mean and standard deviation were calculated for the data presented in the figure. The results are presented in Table 1.

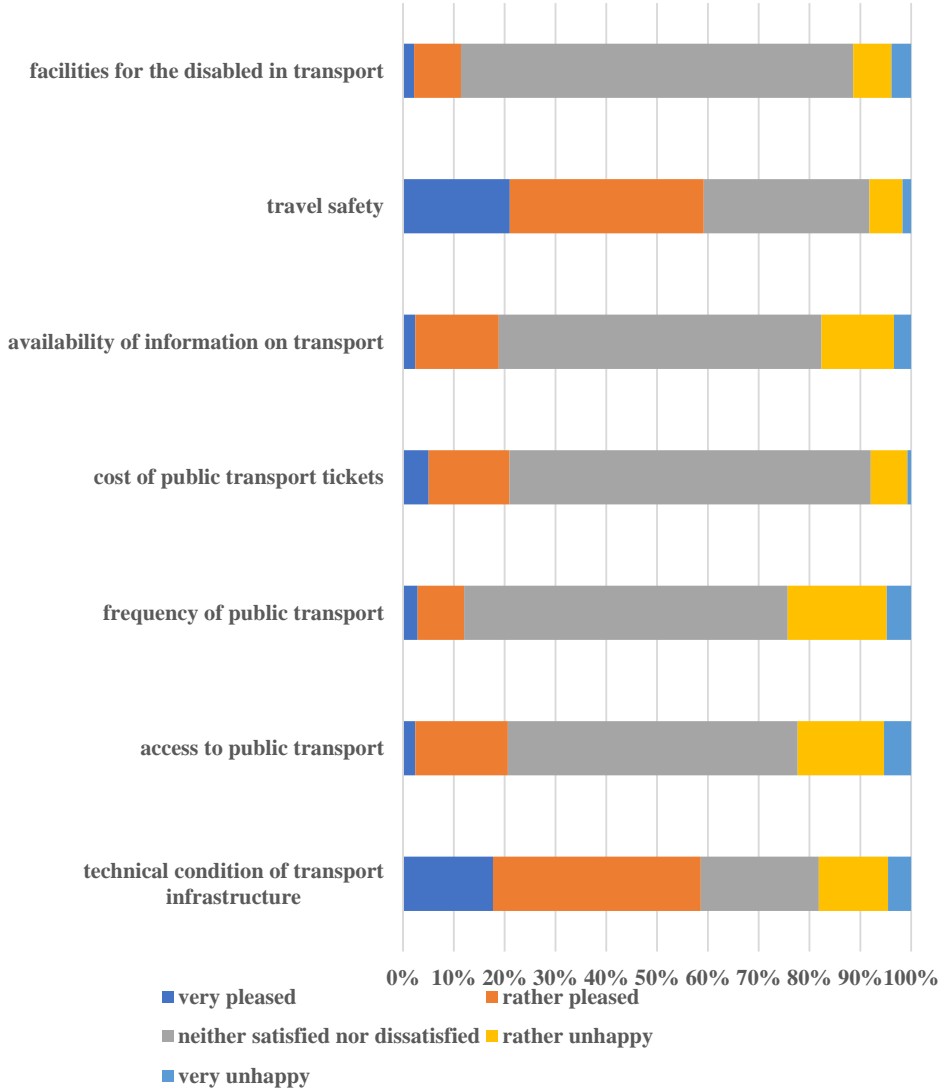

**Figure 1.** The degree of mobility problems of tourists in Hajnowka Poviat with the concentration of transport services. Source: own study based on empirical research.

**Table 1.** Evaluation of the condition and problems of infrastructure of Hajnowka Poviat in the opinion of the tourists (median, mode, arithmetic mean, standard deviation).

| Variable | Tourists | | | |
|---|---|---|---|---|
| | Median Value | Mode | Mean | Standard Deviation |
| Technical condition of transport infrastructure | 4 | 4 | 3.54 | 1.07 |
| Access to public transport | 3 | 3 | 2.95 | 0.81 |
| Frequency of public transport services | 3 | 3 | 2.86 | 0.76 |
| Cost of public transport tickets | 3 | 3 | 3.17 | 0.66 |
| Availability of information on transport | 3 | 3 | 3.00 | 0.73 |
| Travel safety | 4 | 4 | 3.70 | 0.93 |
| Facilities for the disabled in public transport | 3 | 3 | 2.98 | 0.64 |

*Source*: own study based on empirical research.

In terms of the technical condition of infrastructure, Polish respondents highlighted a need to improve the technical condition of roads, including an increase in the number of parking spaces. Tourists highlighted a need for more bike paths, including more parking spaces for bicycles.

The surveys conducted in Russia show that more than half of the respondents (65%) are willing to come to the territory of Zaonezhye to an equipped paid car park with all amenities (catering facility, rubbish collection, toilet, etc.). The highest rating for accessibility in Zaonezhye was given to cashless payment services in shops, petrol stations, etc. (the average score for this service was 7.86). The lowest rating was given to catering services (average rating of 5.24). The overwhelming majority of respondents (88%) met their expectations after visiting the Zaonezhye territory. In addition, the local population is actively involved in the development of tourism infrastructure. During the monitoring period of the MARA project, four new guesthouses (reconstruction of pre-existing historical houses from the 19th–20th century) were opened in the Kizhi skerries area. The local people are looking for options to keep guests in the territory by providing a variety of services. The population is particularly active in the Velikaya Guba area; the year-round transport accessibility of the mainland allows for a diverse range of offers and services. However, there is a problem with finding employees in the hospitality field; as a rule, local residents have no vocational training and young people are not interested in permanent employment in rural areas. The survey shows that more than half of the respondents (65.1%) are willing to come to the territory of Zaonezhye to an equipped paid car park with all amenities (catering facility, rubbish collection, toilet, etc.), while 22% said no.

The results show that the main problem related to tourist mobility diagnosed in both regions concerns road infrastructure. However, the detailed data show some differences mainly due to climatic and economic-political conditions.

## 4. Conclusions

When it comes to the main objective of the research, which was to identify the mobility problems of a tourist destination in remote areas of natural value, it should be considered that this objective has been achieved. Furthermore, the assumption (Hypothesis 1) that road transport is the most popular form of travel in remote areas with natural assets was verified positively, as more than 90% of respondents used this form of transport.

Nevertheless, there is a significant discrepancy in the scope of research conducted and the research material obtained. Namely, the research conducted in the Russian region was significantly limited, due to the political and social conditions and the resulting research needs. The coronavirus pandemic has been a significant obstacle to more extensive research in each region. Despite these obstacles, the results obtained should be considered representative for remote natural value tourism destinations characterised by accidentally valuable assets.

The problems encountered by both groups of tourists in terms of mobility service provision and transport infrastructure represent a gap. Analysis of the results of the survey of tourists travelling in the Hajnowka and Zaonezhye areas in terms of problems related to transport provision and infrastructure indicates the following problems in terms of road infrastructure:

- The car is the absolute dominant means of transportation in both regions, making it necessary to adapt infrastructure to the needs of users;
- The need to improve the technical condition of roads;
- The increase in the number of parking spaces for cars;
- The increase in the number of bike paths in the Hajnowka Poviat;
- The main problems in Zaonezhye are related to roadside services for tourists (the need for free equipped parking areas, campsites, toilets, shops and cafes along the road and availability of waste disposal services).

The apparent differences between the needs of tourists from both regions in terms of detailed infrastructure elements are due to the level of existing infrastructure. Given the

demand for the Zaonezhye area by tourists (as evidenced by the summer of 2020), the many social problems of local residents, the poor development of infrastructure (poor quality of roads, lack of gas stations, power cuts, lack of catering facilities, problems with berths, etc.), it is necessary to consider the development of a separate programme for the development of Zaonezhye.

The research showed that it is possible to formulate some recommendations that are common to both regions, which are as follows:

- Development of information resources that can provide tourists with adequate and timely information concerning transport possibilities and means;
- Development of pedestrian and bike paths and infrastructure.

In addition, respondents visiting the Hajnowka Poviat indicated the need to improve travel conditions for people with disabilities, which should be considered as one of the priorities in upcoming road and transport investments. The current range of transport services in the Hajnowka Poviat indicates that the main problem lies in the frequency of public transport. To a lesser extent, the problem lies in the state of the transport infrastructure, the accessibility of public transport and the availability of information on public transport. The cost of public transport tickets and the safety of travel are rated strongly positive. Promoting a model of private car use by people travelling together is one of the best options.

When formulating practical recommendations for entrepreneurs and state and local authorities, the geopolitical specificities of both regions must be taken into account. The Russian region requires the development of purely tourist infrastructure, mainly accommodation and service services. Based on the analysis, the following recommendations can be considered to improve the mobility and accessibility situation in Zaonezhye:

- The local population should be actively involved in the development of services in passenger and freight transport, excursion services and hospitality;
- The Kizhi Museum should realise plans to create a visit centre at Oyatevshchina and a small multifunctional tourist complex outside of it;
- Car drivers wishing to visit the island should be able to leave their cars there, have lunch and, if they wish, spend the night. It should have a capacity for up to 120 guests and should include parking areas, a café, facilities and an area for camping;
- Road infrastructure development, including road construction, parking places, petrol stations, etc.;
- Reconstruction/construction of the berths;
- Change in helicopter ticket sales' system, with a possibility to buy tickets online.

The added value of the research is the research itself and the research procedure with regard to remote regions with valuable natural assets, as these destinations, due to their uniqueness, should increase the number of incoming tourists by improving accessibility to attractions, but without increasing the negative impact of tourism on these assets. The results obtained, therefore, have both theoretical and practical value. The results of the study coincide with the opinions of other researchers, for example Zamparini and Vergori (2021) and Coppola et al. (2020) regarding transport policies for tourist mobility, which should be synchronised as much as possible and should follow the principles of sustainable tourism, especially one that is environmentally friendly.

The indicated problems in the field of mobility constitute a significant limitation of tourist traffic in both destinations; therefore, the search for new solutions should be preceded by more detailed scientific research aimed at searching for new and also innovative solutions. This recommendation applies mainly to the Russian region, where the presented research was conducted for the first time.

## 5. Limitations

There are some limitations to this study. Firstly, this study investigated problems related to tourist travel behaviour from an environmental and organisational perspective.

However, due to space limitations, this study only considered some of the problems. Future research would need to consider the deeper problems and identify the two groups of respondents based on income level, place of residence and other. The selection of respondents, at the beginning of the study, was random (every tenth person in face-to-face surveys). Unfortunately, this certainty does not exist in the case of Internet-based surveys, especially those conducted in Russia. Changes in research technique were forced by the pandemic. In the case of Poland, there were relatively few cases, as it was only a matter of completing the study, which started in 2019. In contrast, the other partners, including Russia, struggled by starting their research a little later, which was influenced, among other things, by the tourist season, which in Poland occurs in summer and in Russia in winter. Due to this situation, the research continued online, which may have distorted the results, including the profile of a tourist in Russia. It is currently difficult to trace the profile of respondents, as it would be necessary to complete the data on respondents in Russia and then provide a comparison in a table. At present, the war triggered by Russia in Ukraine and the embargo imposed by many democratic countries on the Russian aggressor are hindering this. Secondly, other potential variables could be included in future studies, for example, tourism and transport company competition and regional cooperation.

**Funding:** The research has been financed by the Interreg Baltic Sea Region Program, contract number 100#, within the project of MARA Mobility and Accessibility in Rural Areas—New approaches for developing mobility concepts in remote areas.

**Institutional Review Board Statement:** Not applicable.

**Informed Consent Statement:** Not applicable.

**Data Availability Statement:** Not applicable.

**Conflicts of Interest:** The author declares no conflict of interest. The funders had no role in the design of the study; in the collection, analyses, or interpretation of data; in the writing of the manuscript, or in the decision to publish the results.

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
