# Peer review of "Problems of Tourist Mobility in Remote Areas of Natural Value—The Case of the Hajnowka Poviat in Poland and the Zaoneshye Region in Russia"

_economies, doi:10.3390/economies10090212_

Round 1

Author Response

.

Reviewer 2 Report

The paper is designed as a comparative study, but neither the methodological approach nor the situation is comparable as outlined by the authors them self. The only thing in common is the location in a so "called" rural areas, which is by the way not clearly defined by the authors. In general the paper lacks a descent detail level e.g. on the sites presented the reader does not get detailed information on the accessibility or the infrastructure at site. One could say the same on the description of the methodology in the case of Karelia not even a number of cases is presented, there is no reflection on the methodological approach: e.g. might the 2-years timespan of collecting the data and the different methods have any impact and in case which one.? What about specific literature on the topic? There are tons of high impact articles on public transport in the context of leisure and tourism (Dickinson, Lumsdon, Guiver, Page, Gronau etc.). The literature remains on a textbook-level while neglecting the high-ranked journal articles.

Beyond the low level of academic rigor, there is also no real added value when it comes to the results. Not even mentioning that the recommendations are not based upon the empirical study.  I am sorry for the clear words, but there is nothing to euphemize.

Author Response

.

Reviewer 3 Report

The paper is interesting, but the added value and scientific relevance are limited; the paper does not bring innovative theoretical or methodological contributions; except for the contribution to the knowledge of specific aspects regarding the tourist practices in the two analyzed areas and of the policy recommendations, the paper does not add a scientific value at the current stage of the art. Also, the paper would probably gain in coherence if the author clearly highlighted the specific objectives, hypotesis and the added value, while the author himself acknowledges that the results have limits considering the general aim of the paper.

Overall, the paper is well written, in a clear, coherent style.

Author Response

.

Reviewer 4 Report

Dear authors,

Thank you for sharing your research conducted in Poviat Hajnowka in Poland and Zaoneshye region in Russia. The study has potential, but there are major concerns that need to be addressed. My biggest concern is that the scientific contribution of the study is lacking. The novelty and research gap aspect was not clarified in the introduction or conclusions. Other issues are:

-   Introduction - it should contain the main purpose and the scientific contribution must be made clear (as well as in the conclusion). It is necessary to identify and highlight the gap in the literature that the authors are trying to fill. Case studies should be based on theory, with a well-explained rationale for the study that makes clear the lessons learned (i.e., what lessons could be applied to both theory and other similar destinations).

-         Subchapters 1.1. and 1.2. should form a new chapter, not the introduction. In addition, the general information about the project MARA takes up a lot of space in the introduction and is not directly related or critical to the submitted manuscript.

-         A literature review is missing

-     Study locations - it would be advisable to provide information on tourist arrivals in the regions where the study is conducted.

-         Much more information is needed on the methodology, questionnaire, and survey itself (i.e., how respondents were approached, how the sample was defined, etc.).

-     It is unclear who the respondents are - are they those who visited both sites? The author(s) state that in the second phase some of the respondents were local residents - why? - This should be explained.

-      The profile of respondents is difficult to follow - it would be better if it was presented in a table and separately for the Polish and Russian samples.

-         The results are also not presented very clearly. It would be more valuable if the attitudes for the two groups of respondents were presented separately and then compared to see if there are statistically significant differences between the two groups.

-   Conclusions: Stronger discussion of the results in the context of the previous literature is needed. Research limitations should be included and explained, as well as recommendations for future research.

I hope you find the above comments helpful as you revise your paper.

Good luck!

Author Response

.

Round 2

Reviewer 2 Report

The revision has clearly improved the paper!

Reviewer 4 Report

Dear authors,

Thank you for your revised version. I have just few minor suggestions:

-         at the end of the p. 3 and 4 there is a repetition of a following text:

o   from economic centres;

o   peripherality;

o   mobility and accessibility problems;

o   tourist attractiveness consisting of valuable natural assets

-         The results are still difficult to follow, as they are not presented very clearly. It would be more valuable if the attitudes for the two groups of respondents were presented in the table/tables.